# A versatile high-throughput assay based on 3D ring-shaped cardiac tissues generated from human induced pluripotent stem cell-derived cardiomyocytes

Magali Seguret[1†], Patricia Davidson[2], Stijn Robben[2], Charlène Jouve[1], Celine Pereira[1], Quitterie Lelong[1], Lucille Deshayes[1], Cyril Cerveau[2], Maël Le Berre[2], Rita S Rodrigues Ribeiro[2]*, Jean-Sébastien Hulot[1]*

[1]Université de Paris Cité, PARCC, INSERM, Paris, France; [2]4Dcell, Montreuil, France

*For correspondence:
ana-rita.ribeiro@4dcell.com (RSRR);
jean-sebastien.hulot@inserm.fr (J-SH)

Present address: †Paris Cardiovascular Research Center, Paris, France

**Abstract** We developed a 96-well plate assay which allows fast, reproducible, and high-throughput generation of 3D cardiac rings around a deformable optically transparent hydrogel (polyethylene glycol [PEG]) pillar of known stiffness. Human induced pluripotent stem cell-derived cardiomyocytes, mixed with normal human adult dermal fibroblasts in an optimized 3:1 ratio, self-organized to form ring-shaped cardiac constructs. Immunostaining showed that the fibroblasts form a basal layer in contact with the glass, stabilizing the muscular fiber above. Tissues started contracting around the pillar at D1 and their fractional shortening increased until D7, reaching a plateau at 25±1%, that was maintained up to 14 days. The average stress, calculated from the compaction of the central pillar during contractions, was 1.4±0.4 mN/mm². The cardiac constructs recapitulated expected inotropic responses to calcium and various drugs (isoproterenol, verapamil) as well as the arrhythmogenic effects of dofetilide. This versatile high-throughput assay allows multiple in situ mechanical and structural readouts.

## eLife assessment

This paper reports a **valuable** platform for cardiac tissue cultivation. The throughput, consistency of the tissue, and the potential integration of high-throughput automation are an advantage over other approaches. The tissues and the platform are validated using appropriate methodology to provide **convincing** evidence of the tissue cultivation capability.

## Introduction

Cardiac tissue engineering (CTE) aims to generate in vitro cell constructs that recapitulate the intricate structural and functional properties of the native myocardium (*van der Velden et al., 2022*). As the heart lacks regenerative capacities, CTE emerged in the field of regenerative medicine with the objective of producing cardiac grafts that can be implanted in patients with heart failure (*Madonna et al., 2019*; *Jabbour et al., 2021*; *Eschenhagen et al., 2022*). In parallel, the rapid development of biomaterials and microfabrication techniques combined with progress in pluripotent stem cell biology have enabled the generation of miniature engineered heart tissues (EHTs) used as 'hearts-on-chips' or cardiac organoids models. While these EHTs recapitulate a limited number of functions in vitro, they have provided essential platforms for cardiac disease modeling (*Filippo Buono et al., 2020*; *Cashman*

*et al., 2016*; *Williams et al., 2021*; *Stillitano et al., 2016*; *Bliley et al., 2021*; *Goldfracht et al., 2019*; *Richards et al., 2020*) and drug development (*Beauchamp et al., 2015*; *Polonchuk et al., 2017*; *Zhao et al., 2019*; *Mannhardt et al., 2020*), reducing the need for animal studies.

Different combinations of cells, biomaterials, and scaffolds have successfully generated 3D EHTs (*Seguret et al., 2021*; *Zhuang et al., 2022*) with various geometries including spheroids (*Beauchamp et al., 2015*; *Polonchuk et al., 2017*; *Giacomelli et al., 2020*; *Richards et al., 2020*; *Hofbauer et al., 2021*; *Lewis-Israeli et al., 2021*), cardiac strips (*Hansen et al., 2010*; *Legant et al., 2009*; *Turnbull et al., 2014*; *Mannhardt et al., 2016*; *Nunes et al., 2013*; *Zhao et al., 2019*), circular bundles (*Goldfracht et al., 2020*; *Tiburcy et al., 2017*; *Li et al., 2020*), myocardium-like sheets (*Shadrin et al., 2017*), or ventricle-like chambers (*MacQueen et al., 2018*; *Li et al., 2018*; *Lee et al., 2019*). The EHTs are optimally formed by mixing human induced pluripotent stem cells (hiPSCs)-derived cardiomyocytes, fibroblasts, and/or other cell types such as endothelial cells to more closely mimic the structure and cellular complexity of native myocardium (*Giacomelli et al., 2020*; *Saini et al., 2015*). However, a major obstacle in the development of EHTs is the inevitable trade-off between the need for miniaturization to increase throughput (i.e. several organoids in each well of a 96- or 384-well plate) and the biological complexity of the tissues (*Cho et al., 2022*). The first EHTs were generated in small batches, using pre-designed molds, and having a millimetric size (*Boudou et al., 2012*; *Turnbull et al., 2014*; *Legant et al., 2009*; *Mannhardt et al., 2016*). Since then, two different avenues of development have emerged. On the one hand, to optimize throughput, more straightforward, low complexity assays such as spheroids were developed to obtain smaller tissues (*Beauchamp et al., 2015*; *Polonchuk et al., 2017*). These multicellular constructs are valuable models to study drug responses and the human heart micro-environment but they lack the geometric intricacy of a native cardiac tissue. On the other hand, the search for higher tissue complexity and maturity led to the development of chambered organoids (*MacQueen et al., 2018*; *Li et al., 2018*; *Lee et al., 2019*). These constructs require the use of complex techniques such as bioprinting or the use of bioreactors, and tissues are generated one at a time, making these approaches time-consuming and impractical on a large scale. Therefore, the next step in tissue engineering needs to be directed toward high-throughput physiologically relevant assays that are simple and straightforward for the end-user. Scaling down the tissue size to increase throughput, while controlling their geometry, represents a major engineering challenge. Moreover in actual designs, many EHTs are attached on static posts that can impact tissue formation, result in isometric contraction, and can interfere with optical acquisitions when made with opaque material.

Here, we report the development of a novel assay which allows fast, reproducible and high-throughput generation of 3D cardiac rings in a 96-well plate using hiPSC-derived cardiomyocytes. The design allows the formation of multiple ring-shaped cardiac tissues in a well. The EHTs form around a central pillar made from an optically transparent and deformable polymer that enables in situ monitoring of the tissue contraction with simultaneous measurements of force generation.

## Results

### Design and characterization of the molds

3D steel molds were designed to obtain a structure on which cells can be directed to ring-shaped cavities during sedimentation (*Figure 1A*). The mold fits in the well of a 96-well plate and consists of 21 identical conical structures with a central hole (*Figure 1A*, left). The molds obtained from the manufacturer were imaged by scanning electron microscopy and found to correspond to the intended design (*Figure 1A*, right). Casts of the mold were created in polyethylene glycol (PEG) to check the shape of the inner cavities of the mold, but the resulting structures were too soft to maintain their shape in air, and were transparent in water and therefore could not be imaged. More rigid polydimethylsiloxane (PDMS) was cast from the mold to verify the shape of the resulting structures (*Figure 1B*). These showed that the interior profile of the cavity was shaped in a reversed hourglass, as expected.

### Generation and characterization of the gels

3D gel structures were obtained by adding a drop of PEG gel solution on top of the molds, inverting these onto a glass coverslip, polymerizing the gel and attaching the resulting sample on a 96-well plate. To determine the stiffness of the gels, a sample of the aqueous PEG solution was polymerized. A capillary was placed perpendicularly to the edge of the gel and negative pressure was applied to

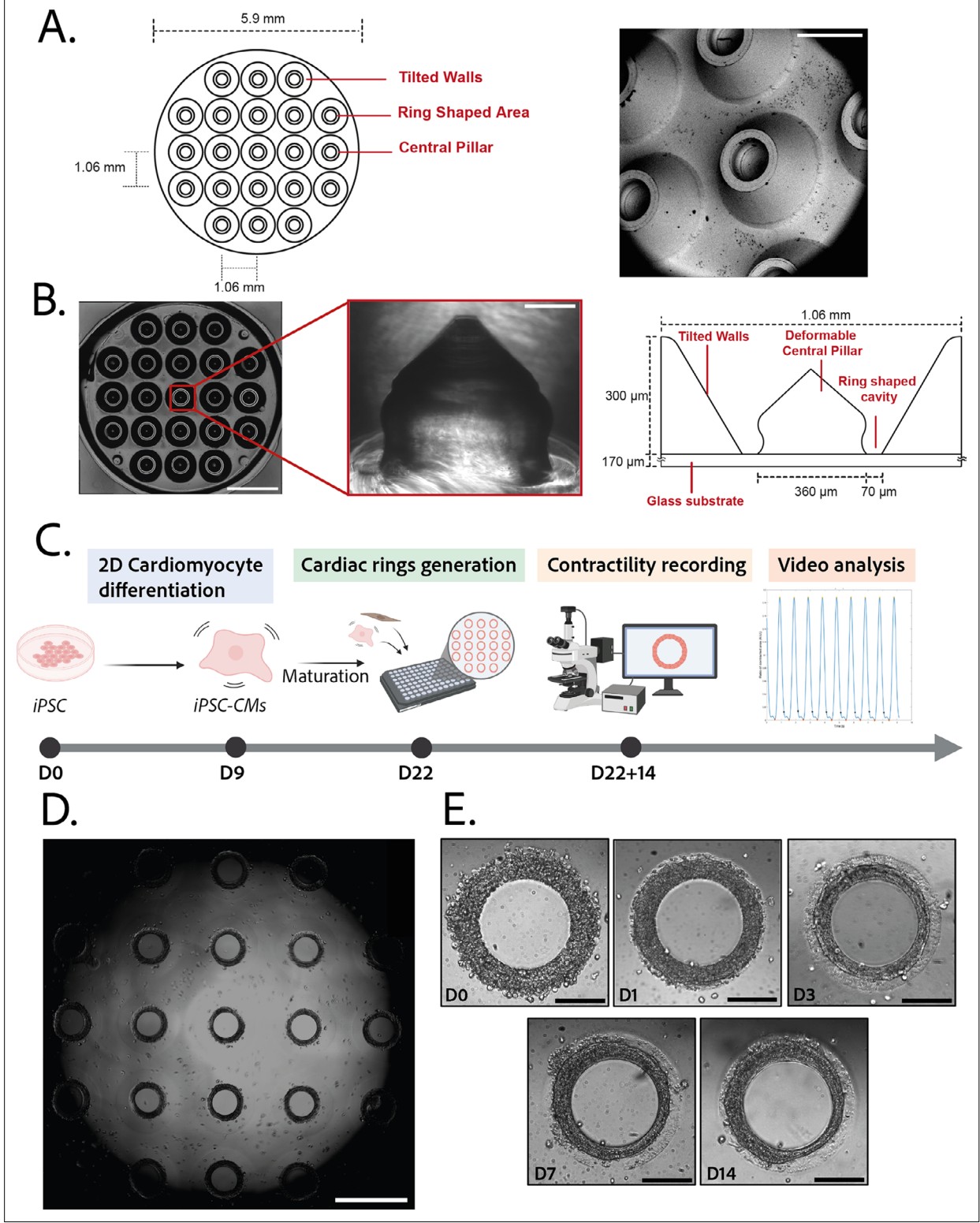

**Figure 1.** Description of the mold and seeding procedure. (**A**) Micromachined stainless steel mold used to shape the gel. Design (left) and scanning electron microscopy image (right) of the mold. Scale bar: 500 μm. (**B**) Molded polyethylene glycol (PEG) gel (left – scale bar: 2 mm) and zoomed view of a pillar from its polydimethylsiloxane (PDMS) replica (middle – scale bar: 100 μm). Design and size of the pillar (right). (**C**) Timeline of the seeding procedure. (**D**) Cardiac rings in a well 1 day after seeding. Images stitched with ImageJ plugin. Scale bar: 1 mm. (**E**) Representative compaction of a ring with time after seeding (from day 0 to day 14), in brightfield. Scale bars: 200 μm.

*Figure 1 continued on next page*

*Figure 1 continued*

The online version of this article includes the following video and figure supplement(s) for figure 1:

**Figure supplement 1.** Measurement of the Young's modulus of polyethylene glycol (PEG) gel.

**Figure supplement 2.** Measurement of the efficiency of differentiation the induced pluripotent stem cells (iPSCs) into cardiomyocytes.

**Figure 1—video 1.** Multiple rings beating at D14 – brightfield imaging ×4 magnification.

https://elifesciences.org/articles/87739/figures#fig1video1

the gel through the capillary. The deformation of the gel into the capillary was measured and used to determine the Young's modulus of the gel (*Figure 1—figure supplement 1*). The values obtained (11.4±0.5 kPa) are within the physiological range for human cardiac tissue (*Gershlak et al., 2013*).

## Generation of ring-shaped cardiac tissues

The efficiency of the differentiation of iPSCs into cardiomyocytes was in average of 96.9±0.8% (*Figure 1—figure supplement 1*).Cardiomyocytes derived from iPSCs (iPSC-CMs) with fibroblasts were seeded on the gels. After centrifugation, the cells sedimented into the ring-shaped cavities (*Figure 1D*) and compacted around the pillars to rapidly form ring-shaped cardiac constructs (*Figure 1D and E*). Initial tests with cardiomyocytes resulted in cell cluster formation and malformation of the rings (*Figure 2A*). To stabilize the tissues, iPSC-CMs were mixed with fibroblasts and different ratios of hiPSC-CMs to fibroblasts were tested in order to obtain the highest number of full stable cardiac rings with time. Three ratios of hiPSC-CMs:fibroblasts were tested (*Figure 2B*) based on previous reports (*Tiburcy et al., 2017*; *Saini et al., 2015*; *Giacomelli et al., 2020*; *Beauchamp et al., 2020*). The 3:1 hiPSC-CMs:fibroblasts ratio was determined to be optimal as it allowed the generation of the highest number of tissues stable in time, and we could not see any difference in the contractile parameters of the tissues with different fibroblast ratios (see *Figure 2—figure supplement 2*). Therefore, the 3:1 hiPSC-CMs:fibroblasts ratio was used for all of the following experiments. Tissues started beating within 24 hours after seeding (*Figure 1—video 1*). The rings compacted in time as shown in *Figure 1E*. In these optimization experiments, we obtained an average of 10.75±0.48 tissues per well which were stable over time for at least 20 days with the 3:1 hiPSC-CMs:fibroblasts ratio.

## Structure and organization of the engineered cardiac tissues

The 3D organization of the tissues was assessed by 3D immunofluorescence imaging 14 days after seeding. The fibroblasts are concentrated at the base of the ring in contact with the glass (*Figure 2C* – bottom panel), whereas the cardiomyocytes form a compact ring above the fibroblasts around the central pillar (*Figure 2C* – top panel). Troponin T staining evidenced the formation of striated elongated fibers (*Figure 2D*), which are typical of cardiac tissue. The obtained cardiac tissues are 75.5±1.8 μm high and their shape is toric, as shown in the 3D reconstruction (*Figure 2E* and *Figure 2—video 1*). To investigate the kinetics of the fibroblasts-cardiomyocytes segregation, the same immunostainings were performed respectively 1 and 7 days after seeding (see *Figure 2—figure supplement 2*). It appears that 1 day after seeding the fibroblasts are not yet attached, although the cardiac fiber has already started to be formed. Seven days after seeding, fibroblasts are fully spread and attached, and the contractile ring is formed and well aligned.

## Contractility analysis

Tissues started contracting less than 24 hr after seeding, which was visible through the deformation of their central pillar (*Figure 1—video 1*). Each tissue was recorded in brightfield imaging, at a ×10 magnification with a high-speed camera (*Figure 3—video 1*). An in-house Matlab code was developed to quantify the contraction and relaxation phases of the tissues by monitoring the area of the central pillar over time (*Figure 3A*). The beat rate of the tissues first decreased between day 1 and day 3 and increased again to reach 0.58±0.030 Hz on day 14 (*Figure 3B*). The contraction stress developed by the tissues increased from day 1 until day 7 (*Figure 3C*). At day 7, the contractility parameters stabilized with an average stress of 1.4±0.1 mN/mm$^2$ (*Figure 3C*). The corresponding developed strain $\epsilon_A$ was of 24.84±0.92% (*Figure 3—figure supplement 1*). Consistently, the relaxation and

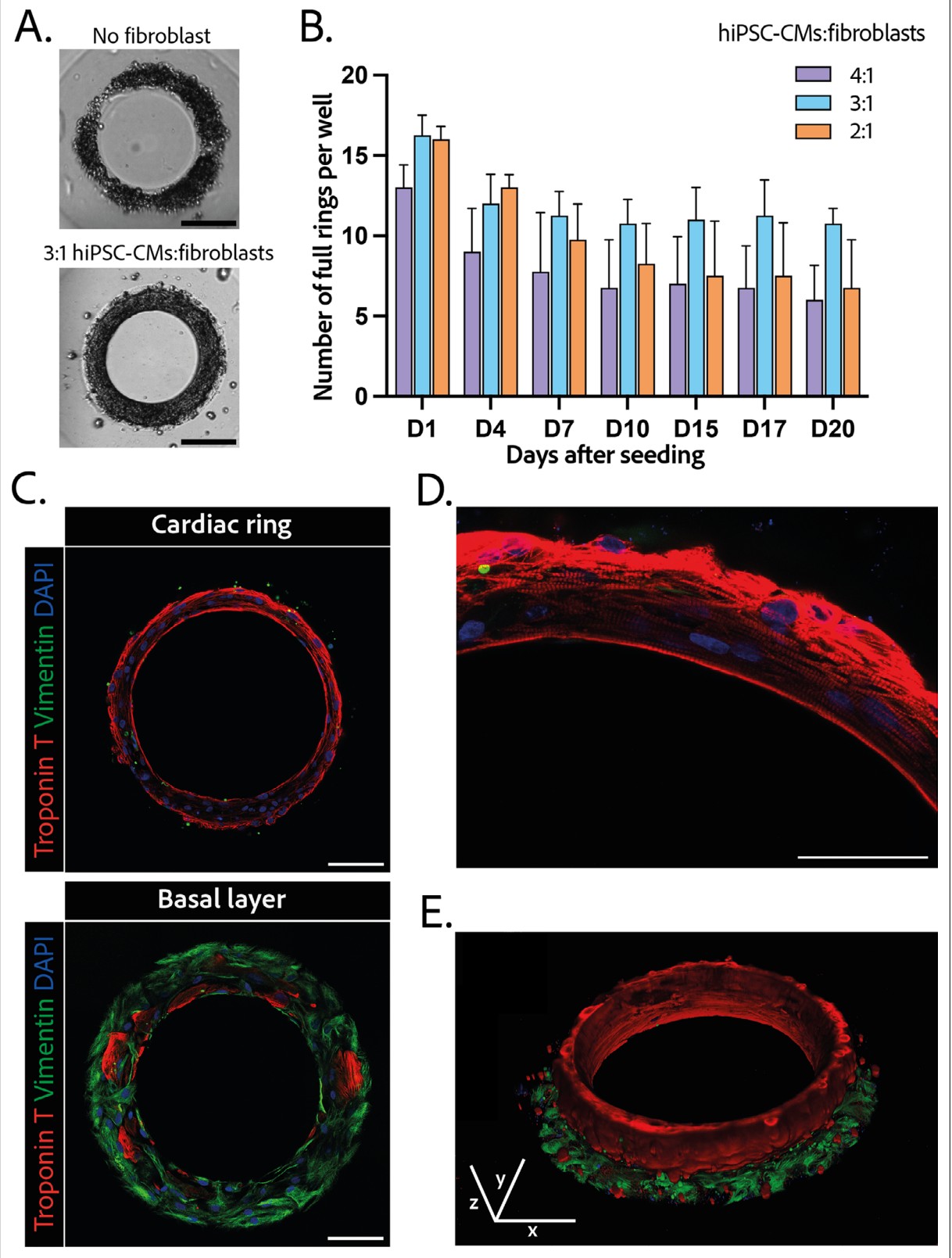

**Figure 2.** Composition and structure of the cardiac rings. (**A**) Example brightfield images of cardiomyocytes seeded with or without fibroblasts 1 day after seeding. Scale bar: 200 µm. (**B**) Number of full cardiac rings per well in time according to the cardiomyocytes derived from human induced pluripotent stem cell (hiPSC-CMs):fibroblasts ratio they contain (4:1, 3:1, or 2:1). Mean ± SD, for n=4 differentiations. (**C**) Confocal imaging of immunostained tissues at different heights: cardiac ring (top panel), and basal layer (bottom panel) of a tissue at ×40 magnification. Vimentin,

*Figure 2 continued on next page*

*Figure 2 continued*

stained in green, corresponds to fibroblasts, troponin T, in red, is specific to cardiomyocytes, and DAPI is in blue. Scale bars: 100 µm. (**D**) Picture of the immunostained contractile fibers at ×63 magnification. Vimentin (green), troponin T (red), and DAPI (blue). Scale bar: 50 µm. (**E**) 3D reconstruction of a ring. x, y, and z scale bars: 100 µm.

The online version of this article includes the following video and figure supplement(s) for figure 2:

**Figure supplement 1.** Contractile parameters at D14 for the different fibroblasts ratios.

**Figure supplement 2.** Magnified regions of rings displayed in *Figure 2C*.

**Figure supplement 3.** Immunostaining of rings at D1 and D7.

**Figure 2—video 1.** 3D reconstruction of a ring.

https://elifesciences.org/articles/87739/figures#fig2video1

contraction speeds also increased during the first 7 days, then reaching a plateau (*Figure 3D and E*). Since long-term culture of iPSC-CMs is gaining interest, we studied the contractile parameters of our rings 28 days after seeding in comparison to their contractile parameters at D14 (see *Figure 3—figure supplement 2*). We found a slight increase for all the parameters, which is significant for the maximum contraction speed. Nevertheless, the data is much more variable and the number of tissues is lower (29 for D14 against 17 for D28). Therefore, we demonstrated that long-term culture of our tissues is possible, however not yet optimized. Hence, the following physiological and pharmacological tests have been done at D14.

## Arrhythmia analysis

The regularity of the rings was first evaluated by plotting each period (time between two contraction peaks) as a function of the following one. We found that the Poincaré plot was fitted with a $y=x$ line with $R^2 = 0.9633$ (*Figure 4A*) and with a narrow confidence interval (CI), showing that the vast majority of the $n$th periods are equal to the following ones, and therefore that the rings beat regularly without spontaneous extra beats. Also, following the study of *Li et al., 2020*, which shows the occurrence of re-entrant waves in the case of circular geometry of tissues, we performed live imaging of a voltage-sensitive fluorescent dye in our tissues and looked for re-entrant waves. We found that, as expected, the tissue depolarization induced a visible increase in fluorescence (see *Figure 4B*). All the pixels were synchronous and we could not detect any sparks or re-entrant waves (see *Figure 4—video 1*). We then developed a Matlab script to analyze the average FluoVolt fluorescence signal together with the contraction over time, which was also obtained from the same videos. We found that the tissues beat regularly, with a depolarization occurring just before the contraction of the ring, causing a large increase in tissue fluorescence (from 0 to 6 $\Delta F/F_0$). We also observed that the repolarization signal is noisy, as it occurs during the time when the tissue deformation is greatest and is probably affected by tissue motion.

## Physiological testing on the engineered cardiac tissues

The response in contractility of the cardiac rings to an increase in the extracellular calcium concentration was assessed by sequential addition of calcium in the medium, starting from a concentration of 0.5 mM up to 3.5 mM (*Figure 5A*). The changes in the contraction stress and the contraction and relaxation speeds with calcium concentration were significant (p<0.0001 for each parameter, non-significant for the beat rate). We observed a stabilization of the beat rate around 0.5 Hz at 2 mM and a positive inotropic response as the contraction amplitude increased by 600% with a corresponding $EC_{50}$ of 1.346 mM with a CI of [1.159;1.535] mM.

## Pharmacological testing on the engineered cardiac tissues

We next evaluated the dose-response to increasing concentrations of several cardiotropic drugs. The changes in the beating parameters with verapamil concentration were found to be significant (p=0.0053 for the beat rate and p<0.0001 for the other parameters). The addition of verapamil, an L-type calcium channel inhibitor, induced a negative inotropic effect until the complete stop of contraction at 10 µM, as expected (*Figure 5B*). The $IC_{50}$ was estimated to be 0.677 µM, CI=[4.278;12.81]·$10^{-7}$ M. The dose-response to the β-adrenergic receptor agonist, isoproterenol, was tested. We observed a trend toward increase of the contraction amplitude, as well as a significant increase in the contraction

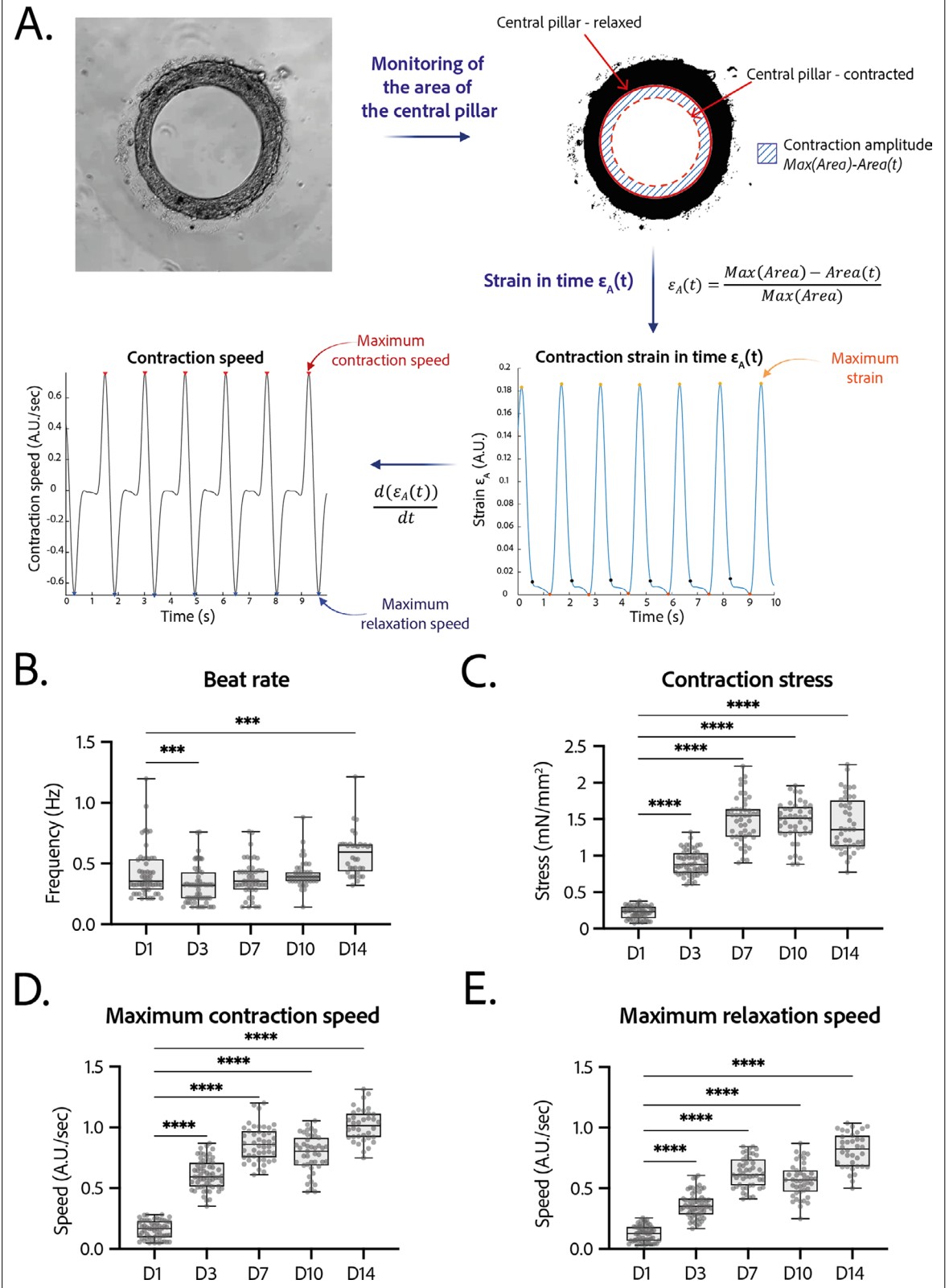

**Figure 3.** Contractility analysis of the tissues. (**A**) Principle of the in-house Matlab code used for contractility analysis: detection of the central pillar and monitoring of the evolution of its area in time, calculation and plot of the strain $\epsilon_A$ in time (ratio between the contraction amplitude in time and the maximum area of the central pillar). Representative plots of the strain in time and its derivative in time, for a tissue at day 14. (**B–E**) Evolution of beating parameters through time after seeding at days 1, 3, 7, 10, and 14. The changes of all the parameters through time are significant (p<0.0001 – ANOVA for

*Figure 3 continued on next page*

*Figure 3 continued*

repeated measures – D1: n=57, D3: n=59 D7: n=47, D10: n=43, D14: n=36 tissues, from three differentiations). Beating parameters at each time point are compared to their value at day 1. (**B**) Evolution of beat rate through time after seeding (***: $p<0.002$). (**C**) Evolution of contraction stress through time after seeding (****: $p<0.0001$). (**D**) Evolution of maximum contraction speed through time after seeding (****: $p<0.0001$). (**E**) Evolution of maximum relaxation speed through time after seeding (****: $p<0.0001$).

The online version of this article includes the following video and figure supplement(s) for figure 3:

**Figure supplement 1.** Strain $\epsilon_A$ developed by the rings at D14.

**Figure supplement 2.** Contractile parameters of rings at D28.

**Figure 3—video 1.** Contractility of a ring at day 14.

https://elifesciences.org/articles/87739/figures#fig3video1

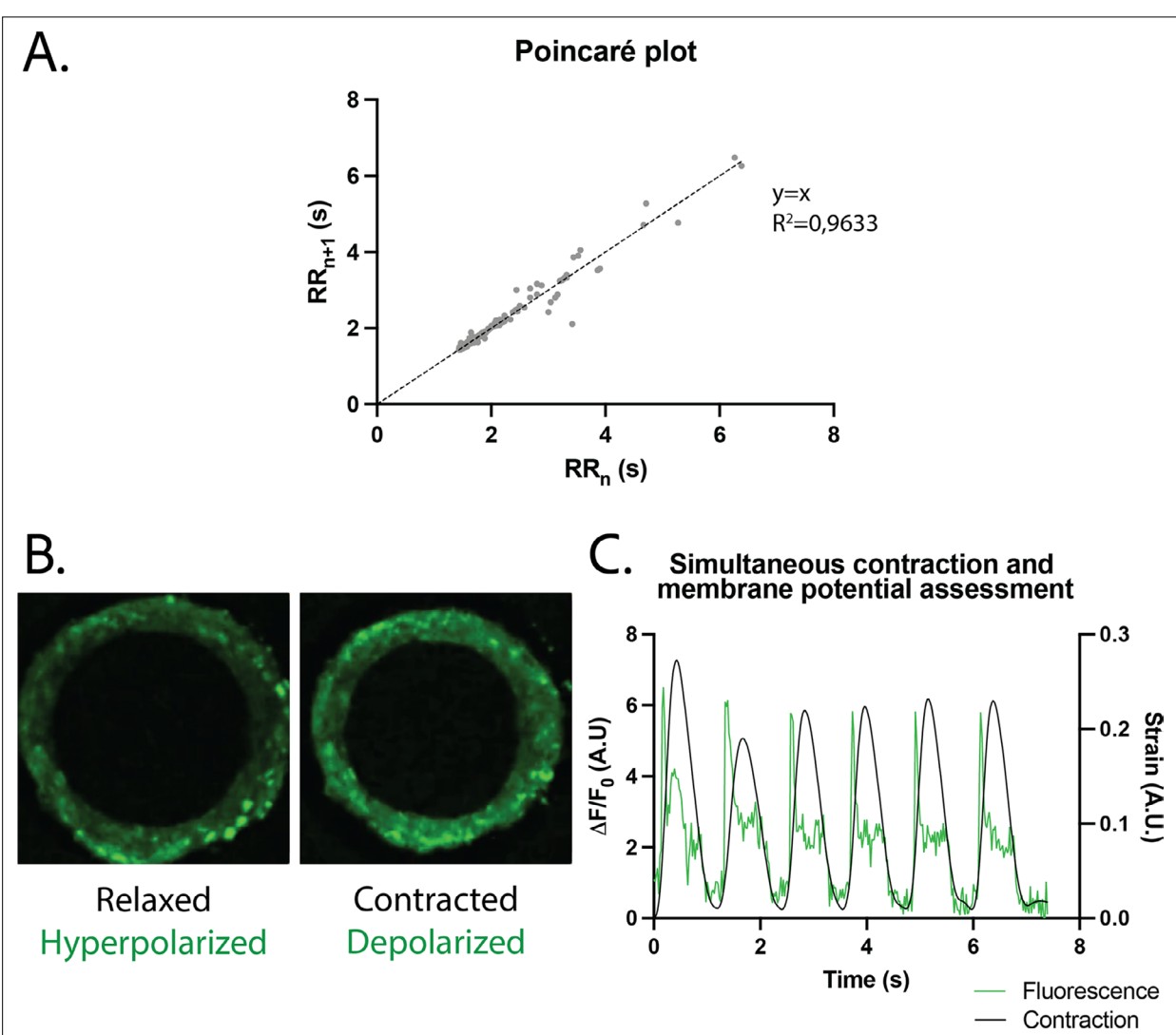

**Figure 4.** Study of arrhythmia in ring-shaped cardiac tissues. (**A**) Poincaré plot for tissues at D14. n=24 tissues from three differentiations. (**B**) Tissue with FluoVolt dye in its relaxed/hyperpolarized state and contracted/depolarized state. (**C**) Representative plot of contraction strain (black) and fluorescence signal (green) of a tissue derived from video analysis of the FluoVolt dye.

The online version of this article includes the following video for figure 4:

**Figure 4—video 1.** FluoVolt live imaging of a ring at D15.

https://elifesciences.org/articles/87739/figures#fig4video1

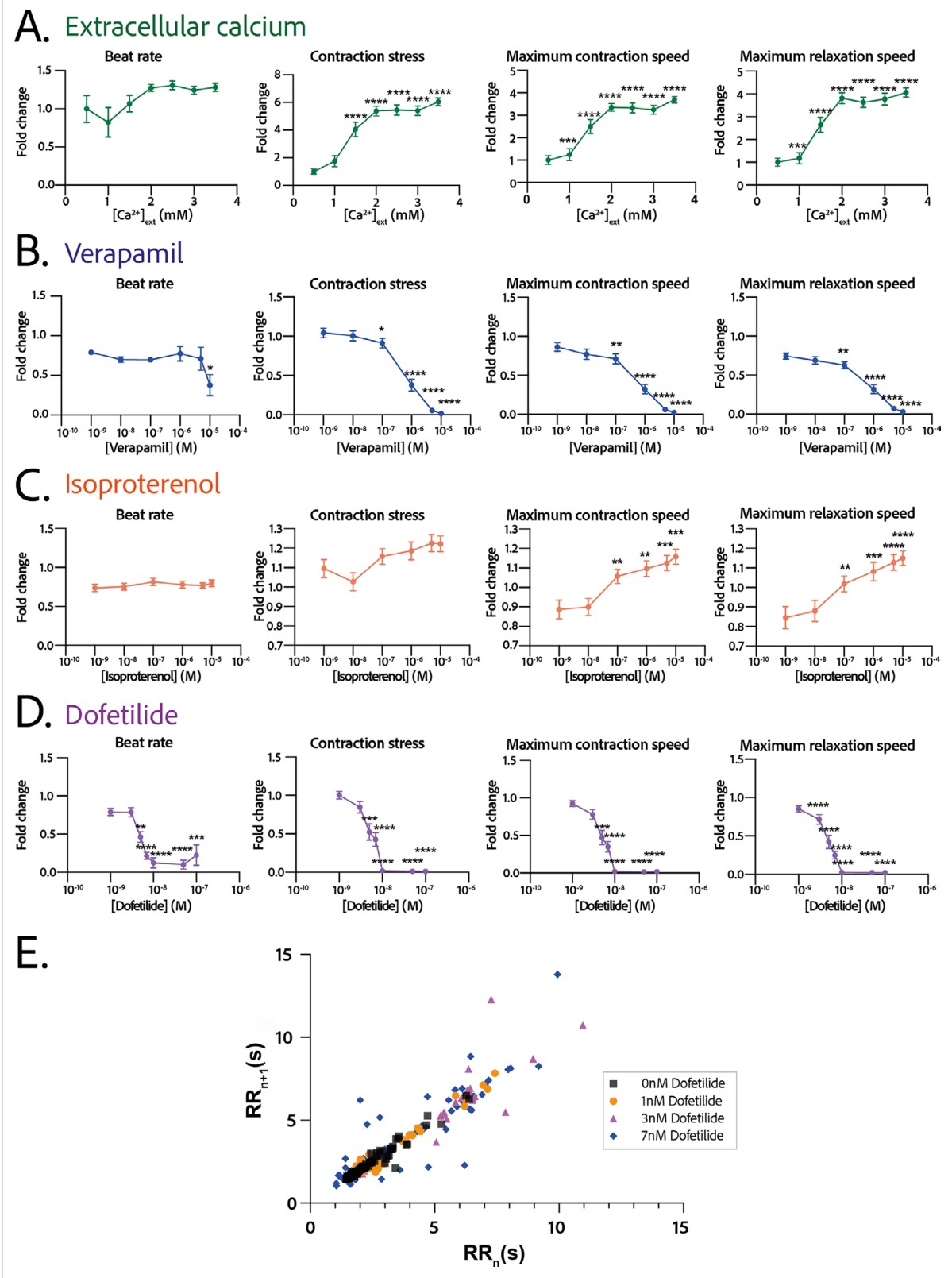

**Figure 5.** Physiological and drug testing on the cardiac tissues. Effect of the concentration in extracellular calcium (**A**), verapamil (**B**), isoproterenol (**C**), or dofetilide (**D**) on tissues contractility: beat rate, contraction stress, and the maximum contraction and relaxation speeds. These parameters are expressed as a ratio between their value for each concentration and the value at basal state ($[Ca^{2+}]$=0.5 mM for calcium test and [Drug]=0 M for drug testing). An ANOVA for repeated measures was carried out for each parameter. For each test, the value of each parameter at each concentration

*Figure 5 continued on next page*

*Figure 5 continued*

is compared to its value at the minimal concentration of the drug (respectively $[Ca^{2+}]$=0.5 mM, [verapamil]=$10^{-9}$ M, [isoproterenol]=$10^{-9}$ M, and [dofetilide]=$10^{-9}$ M). (E) Poincaré plot for 0 nM, 1 nM, 3 nM, and 7 nM of dofetilide. For each concentration of each drug, more than 20 tissues from three different concentrations could be analyzed. Data is presented as mean ± standard error mean (SEM). *: p<0.05; **: p<0.01; ***: p<0.001; ****: p<0.0001.

and relaxation speeds (*Figure 5C*), without change in the beat rate. The changes in the contraction stress and the contraction and relaxation speeds with isoproterenol concentration are significant (p=0.0004 for the contraction stress; p<0.0001 for the contraction and relaxation speeds). Finally, dofetilide, which is an inhibitor of the $I_{Kr}$ current, was also tested at different concentrations on the cardiac rings. This drug has been shown to be pro-arrhythmic and to induce arrhythmic events such as Torsades-de-Pointes in patients. The changes in the beat rate, the contraction stress, and the contraction and relaxation speeds with dofetilide concentration were found to be significant (p<0.0001 for all the parameters). As shown in *Figure 5D*, dofetilide significantly reduced the contraction stress and speeds until the complete stop of the contraction of the cardiac tissues for a concentration of 10 nM. Arrhythmic events were noted after adding dofetilide at concentrations of 3 nM and 7 nM. Indeed, the Poincaré diagram for the different dofetilide concentrations (*Figure 5E*) shows an increase in the contraction-to-contraction intervals (equivalent to RR intervals) for some cells at 1 nM of dofetilide and an increasing number of erratic beats with increasing dofetilide concentrations.

## Discussion

The combination of pluripotent stem cells technologies, advanced cardiomyocyte differentiation protocols, and microfabrication methods have already resulted in the generation of engineered human cardiac muscle tissue constructs, with different shapes and that can reproduce some cardiac functions (*Seguret et al., 2021*; *Cho et al., 2022*). However, until now, approaching the complex geometry and structure of the human myocardium required complex techniques, which limit the number of tissues produced, thus hampering their application for high-throughput experiments. In this study, we aimed to address this challenge by developing a novel strategy to create ring-shaped miniature cardiac tissues around a central pillar that is made from an optically transparent and deformable polymer. Our main findings are as follows: (1) This novel strategy enables the fast and reproducible formation of an important number of cardiac tissues (up to 21 per well in a 96-well format) with a limited number of hiPSC-CMs (112,500 per well). (2) In this assay, fibroblasts are essential to generate and maintain the tissue structure, with a 3:1 hiPSC-CMs:fibroblast ratio providing with the best yield of tissue generation. (3) The self-organized ring-shaped cardiac tissues contract and deform the optically transparent central pillar, which can be recorded and analyzed to estimate the contractile force developed by the tissue. (4) These ring-shaped tissues are compatible with live fluorescence imaging and allow the simultaneous measurement of contractile parameters and another readout provided by a fluorescent probe. (5) These EHTs display concordant responses to positive or negative inotropes, and to arrhythmogenic drugs, as required for drug testing applications.

The miniaturized cardiac constructs develop by self-organization of the hiPSC-derived cardiomyocytes and fibroblasts, which are guided by a specific design to acquire a ring shape. The resulting EHTs are in a convenient 96-well plate format and easy to use for a user familiar with cell biology techniques. The 96-well plate format is adaptable to many plate readers and automated platforms. In addition, the optical clarity of the gels and the absence of gel directly under the structures ensures optimal imaging conditions. We showed that the tissues can be fixed, stained by immunofluorescence and imaged in situ at high resolution. Moreover, our EHTs are also be compatible with live cell imaging experiments with fluorescent reporters, such as the FluoVolt voltage-sensitive dye, for which we presented a proof-of-concept. Indeed, we demonstrated that while measuring the depolarization of the cells by imaging the fluorescence over time, we could simultaneously monitor the contraction parameters in situ through the deformation of the pillar. This possibility to simultaneously record two parameters represents a significant asset, as this proof-of-concept can be extended to other fluorescent probes.

The current design of our EHTs consists of up to 21 rings in a single well, thus providing many replicates within a single experiment, even if some of the rings fail to form due to biological variability. This high throughput can compensate for potential intra-batch effects. Moreover, the possibility to

seed several wells at the same time allows to easily study several differentiations, compensating for the variability which is inherent to the differentiation of hiPSCs. The predictable circular shape of the tissues makes them easy to recognize and track using software. Coupled with the known organization of the 21 rings for easy automated imaging, a single well can quickly generate data on many tissues that is quickly and automatically analyzed. The EHTs we described here are thus easy to use, image, manipulate, and obtain readouts from.

We found that the circular geometry of these self-organized cardiac tissues induces a homogeneous distribution of the cardiomyocytes and consequently of the contraction forces around the pillar and in the tissue. Importantly, the presence of the central pillar not only allows resistance to be felt by the tissue, but the control of its stiffness facilitates the calculation of the force developed by the tissue. Our results showed that the tissues develop a fractional shortening (or contraction strain) of about 25%, which is close to the contraction of a human heart (*Cheng et al., 2010*). The corresponding stress was 1.4 mN/mm$^2$ which is in a similar range to other previously reported tissue-engineered cardiac muscle models with hiPSC-CMs in the literature (*Seguret et al., 2021*; *Turnbull et al., 2014*; *Shadrin et al., 2017*; *Zhao et al., 2019*; *Ronaldson-Bouchard et al., 2018*; *Goldfracht et al., 2020*; *Li et al., 2020*). Some studies have reported higher force values (*Ronaldson-Bouchard et al., 2018*) but in tissues that have a much large size than the tissues proposed here. The force generated by the adult human heart muscle, around 44 mM/mm$^2$ for a tissue strip (*Hasenfuss et al., 1991*), however remains significantly higher than the force developed by the currently available platforms. Few cardiac assays can precisely monitor the force exerted by the tissues. Indeed, cardiac cells seeded on micropillars (*Rodriguez et al., 2014*) and gels containing tracking beads (*Dou et al., 2021*; *Feyen et al., 2020*) can provide information on local (subcellular) force exertion. The force measurements do not result from the concerted effort of an organized tissue composed of aligned cardiomyocytes, as proposed by ring-shaped tissues. Tissues attached on two posts (*Turnbull et al., 2014*; *Mannhardt et al., 2016*; *Abilez et al., 2018*) can also provide some information on the force generated by a tissue but the force exertion is not radial as it is in the native heart. Moreover, the presence of the posts can impact the contraction by affecting the force distribution in the tissue and creating potential edge effects in the vicinity of the posts, inducing the formation of v-necks (*Abilez et al., 2018*). Lastly, some other ring-shaped cardiac constructs have to be transferred on posts or on a force transducer to measure their contraction force (*Goldfracht et al., 2020*; *Tiburcy et al., 2017*). Therefore, the easy in situ monitoring of the force generated by our ring-shaped tissues with an image analysis routine represents a significant advantage, as less manipulation improves the degree of standardization. Finally, the stiffness of the central pillar can easily be adapted to model a pathological increase or decrease of the extracellular matrix stiffness.

The EHTs we developed are composed of cardiomyocytes that rely on fibroblasts for their assembly and inter-cellular adhesion. Fibroblasts have been shown to play a prominent role in the heart as they are crucial in the constitution of the supporting extracellular matrix and contribute to cardiomyocyte electrical coupling, conduction system insulation, vascular maintenance, and stress-sensing (*Ivey and Tallquist, 2016*; *Baudino et al., 2006*). Other studies have highlighted the importance of a multicellular approach to generate EHTs (*Giacomelli et al., 2020*; *Saini et al., 2015*; *Amano et al., 2016*; *Caspi et al., 2007*). In our system, the fibroblasts additionally play a role in providing a basement support under the cardiomyocytes, in contact with the glass support of the EHTs. Immunofluorescence imaging showed an organization of the cardiomyocytes parallel to each other, perpendicular to the direction of contraction, with an increased alignment of sarcomeres. Our EHTs mimic the composition and organization of the native myocardial tissue, even if only composed of cardiomyocytes and fibroblasts. Future experiments will determine whether other, less abundant, cell types of the native myocardium, such as endothelial or smooth muscle cells, can further improve heart tissue modeling.

The constructs showed a spontaneous beating in the day following seeding and this activity was maintained over several weeks thus allowing contractile measurements and pharmacological testing in the conditions of auto-pacing. Since *Li et al., 2020*, highlighted the occurrence of re-entrant waves in their circular cardiac tissues, we investigated the arrhythmogenicity of our tissues and performed live imaging of depolarization in the tissues to evaluate the presence of re-entrant waves. We were able to show that our cardiac tissues beat regularly, and we were able to correlate the depolarization with contraction over time. The next step would be to correct the fluorescence profile to account for tissue deformation in order to study tissue repolarization. This would require the development of a

complex pixel-tracking code. The analysis of re-entrant waves is however hampered by a technical limitation. Indeed, it has been shown that the conduction velocity in iPSC-CMs is around 10–20 cm/s, compared to 40–60 cm/s in adult cardiomyocytes (*Ahmed et al., 2020*). Here, we image our tissues at the maximum speed allowed by our camera, which is between 40 and 50 frames/s (fps), which is faster than in *Li et al., 2020*. However, our constructs are smaller, with a perimeter of about 1.13 mm, and therefore, the required camera speed to detect the conduction in tissues should be at least of 120 fps if we consider a conduction speed of 15 cm/s. Hence we were not able to detect conduction in our tissues, nor re-entrant waves. This first study is nevertheless a proof-of-concept for live imaging in miniature ring-shaped tissues, already allowing us to correlate the depolarization peaks with contraction peaks thus reflecting the excitation-contraction coupling efficiency in the tissues. Additional efforts, especially with a faster camera, will be needed to more accurately analyze tissue conduction and detect re-entrant waves. As a proof-of-concept, we also studied the effects of well-established negative and positive inotropes and showed the ability of our platform to reproduce the anticipated effects. We observed that our tissues presented a positive inotropic response to increasing extracellular calcium concentration with a corresponding $EC_{50}$ which is in agreement with the values reported in the literature for EHTs made of hiPSC-CM, which are between 0.4 mM and 1.8 mM for the $EC_{50}$ (*Feric and Radisic, 2016*; *Schaaf et al., 2011*; *Streckfuss-Bömeke et al., 2013*; *Mannhardt et al., 2016*; *Turnbull et al., 2014*; *Goldfracht et al., 2020*). The negative inotropic response to increasing concentrations of verapamil of our constructs was also described in the literature and our $IC_{50}$ value is also in the range of reported values in other EHTs: *Turnbull et al., 2014*, found $IC_{50}$=0.61 μM and other papers range from 0.3 μM to 0.6 μM (*Mannhardt et al., 2016*; *Thavandiran, 2019*). The β-adrenergic stimulation of the tissues with increasing concentrations of isoproterenol induced a trend toward increase in the contraction force and significantly increased contraction and relaxation speeds without change in the beat rate. This result may be surprising, as many articles report a positive chronotropic behavior for spontaneous hiPSC-CMs, while the inotropic response is less often highlighted (*de Lange et al., 2021*; *Chen et al., 2015*; *Ronaldson-Bouchard et al., 2018*). Nevertheless the increase in contraction and relaxation speeds that we observed is consistent with data reported in the literature. Our hypothesis is that this increase in contraction and relaxation speeds induced by isoproterenol is translated, on average in our study, into an increase in contractile force rather than in an increase in contraction frequency. This may depend on the cell line used, and is very well illustrated in *Mannhardt et al., 2020*: of the 10 different cell lines tested in EHTs, all show an increase in contraction and relaxation speeds after isoproterenol administration, but this is translated either into an increase in contractile force (four cell lines) or into a shortening of the beat (three cell lines), and only two cell lines show an increase in both parameters. Indeed, since hiPSC-CMs are immature cardiac cells, it is rare to obtain a positive force-frequency relationship without any maturation medium or mechanical or electrical training. Last, applying increasing dofetilide concentrations on our constructs showed a decrease in the contraction amplitude and contraction and relaxation speeds. This is in line with the literature as *Zhao et al., 2019*, depicted a significant reduction of the action potential amplitude and an increase in action potential durations of the tissues with increasing dofetilide concentration, in line with *Huo et al., 2019*; *Blinova et al., 2017*. Above a concentration of 10 nM, dofetilide shows cardiotoxicity in our tissues as tissues completely stop beating. By assessing the beating regularity by plotting each period as a function of the previous one for doses of 0 nM to 7 nM of dofetilide, we highlighted an increasing number of erratic beats with the increase in dofetilide concentration. Other papers have also underlined the pro-arrhythmic effect of dofetilide on iPSCs-CMs (*Goldfracht et al., 2019*; *Patel et al., 2019*; *Altrocchi et al., 2020*; *Blinova et al., 2017*) and this drug has been shown to induce atrial fibrillation, Torsade-de-Pointes, and heart failure in patients (*Abraham et al., 2015*; *Shantsila et al., 2007*). Therefore, we demonstrated that our platform recapitulated the expected cardiac responses to several inotropes and arrhythmogenic drugs.

While representing a significant improvement, our platform presents some limitations. First, the size of the rings is very small compared to native tissue (400 μm). This size limitation is necessary for the purposes of high-throughput screening, but for applications requiring larger tissues, wider rings can be considered in the future. Second, all measurements were performed under spontaneous pacing as a specific technique should be developed to allow external pacing of the tissues in the 96-well format. This includes electrical field stimulation or an optogenetics system with light stimulation. The precise and regular organization of the rings in the well makes this design very amenable to the introduction

of microelectrodes for pacing, which could also allow the measurement of the action potentials of individual micro-tissues. Nevertheless, our main results show that pharmacological responses, including the detection of the pro-arrhythmogenic effects of dofetilide, were correctly measured under spontaneous beating conditions. Third, the miniaturized format of our platform creates some limitations and challenges regarding some experiments, including tissue stretching. Therefore, the length-tension relationships (Franck-Starling mechanism) were not determined in our tissues.

Overall the novel platform we describe here is a versatile, easy-to-use, and high-throughput tool to generate cardiac rings in a reproducible way and analyze them in situ. This platform will be useful in several cardiac research fields, including disease modeling and pharmacological testing.

## Materials and methods
### Mold design and characterization
Micromachined stainless steel molds were obtained from high-resolution CNC machining according to our CAD designs. The designs consisted of a mold that would be slightly smaller than the footprint of a well of a 96-well plate, composed of 21 structures (see *Figure 1A*). To check the shape of the interior cavities of the mold, a cast of the mold was created out of PDMS (Sylgard 184, Samaro), according to the manufacturer's instructions.

### 3D hydrogel substrate preparation
A solution of 5% wt/wt photopolymerizable PEG was prepared in distilled water. Circular coverslips (16 mm, Paul Marienfeld GmbH, Germany) were silanized to ensure adequate bonding between the PEG and glass. A drop of 10–15 µL of gel was placed on the stainless steel mold, which was then placed in contact with a silanized coverslip. The gel solution was briefly exposed to light of the appropriate wavelength to polymerize the hydrogel. The resulting 3D structured hydrogel, bound to the coverslip, was unmolded and immediately placed in distilled water. The hydrogel was rinsed at least three times with distilled water and incubated in water overnight. For cell experiments, the coverslip coated in a 3D hydrogel was briefly dried and attached to a commercial black polystyrene 96-well plate with an adhesive backing (Grace Bio-Labs, USA). The resulting well was filled with distilled water. Prior cell seeding, the plate was sterilized by UV irradiation for at least 20 min.

### Young's modulus measurements of the molds
The PEG-based hydrogel solutions were prepared and 100 µL of solution was deposited into a ring-shaped mold with an inner diameter of 1 cm positioned on a 16 mm glass coverslip. The solution was polymerized using 365 nm UV light for 1 min at a power of 70 mW/cm$^2$, after which the mold was removed. The coverslip containing the gel was then transferred to the microscope (Leica Camera AG, Germany). A hollow glass capillary with an inner diameter 0.75 mm (World Precision Instruments, USA) was attached to a platform movable in the *xy*-plane which was used to position the capillary right next to the edge of the circular polymerized hydrogel. Once the capillary and the hydrogel made contact, a negative pressure was applied, using the Cobalt autonomous pressure pump (Elvesys, France), creating a small convexity in the surface of the hydrogel. Images of this convexity were taken (×5 magnification) for a pressure ranging from –100 mbar to –200 mbar, with 10 mbar increments (*Figure 1A*). Two measurements were done on two separate gels, resulting in four measurements per gel with 10 datapoints each. The images (and thus the size of the convexities) were analyzed using ImageJ (version 1.53c). The Young's modulus was calculated with the formula $E = \frac{p}{0.872\frac{l}{a} + 0.748(\frac{l}{a})^2}$ according to the literature (*Gandin et al., 2021*).

### Cardiomyocyte differentiation and culture
The SKiPSC-31.3 hiPSC cell line, from a healthy control subject, has been used here and was previously reported (*Galende et al., 2010*). iPSCs were seeded on Matrigel and cultivated in mTESR Plus medium (StemCell Technologies). At 80% confluency, cells were passaged with ReleSR reagent (Stem-Cell Technologies). When hiPSCs reached a 90% confluency, differentiation was carried out according to a protocol adapted from *Garg et al., 2018*. Briefly, on day 0 of differentiation, mTESR Plus was changed to RPMI-1640 (Life Technologies) supplemented with B27 without insulin (RPMI-B27 minus insulin) and 6 µM CHIR-99021 (Abcam). After 48 hr, the medium was changed to RPMI-B27 minus

insulin and then to RPMI-B27 minus insulin with 5 µM IWR-1 (Sigma) for 48 hr. On day 5, the medium was replaced by RPMI-B27 minus insulin, and switched to RPMI-B27 with insulin on day 7. At day 11 a glucose starvation was carried out by replacing the medium with RPMI 1640 without glucose (Life Technologies), supplemented by B27 with insulin for 3 days. Cells were then dissociated with 0.05% trypsin (Life Technologies) and replated in a 12-well plate at a density of $0.3\ 10^6$ cells/cm$^2$ in RPMI-B27 with insulin. The following day, the medium was switched back to RPMI 1640 without glucose (Life Technologies), supplemented by B27 with insulin for 3 days again. These two rounds of glucose starvation have been shown to increase dramatically the percentage of cardiomyocytes obtained (*Sharma et al., 2015*). After day 18, the cells were cultured in RPMI-B27 with insulin and the medium was changed every 2 days.

## Cardiomyocyte characterization

The efficiency of the differentiation was assessed by cardiac troponin T flow cytometry at D21. On day 21 of differentiation, hiPSC-CMs were dissociated by enzymatic digestion (Miltenyi Multi Tissue dissociation kit 3) and stained with the Zombie NIR Fixable Viability Kit (BioLegend). Then, collected cell pellets were fixed and permeabilized using Inside Stain kit (Miltenyi Biotech, 130-090-477) at room temperature for 10 min. Cells were incubated with either APC anti-cardiac troponin T (CTNT) antibody (Miltenyi Biotech; 130-106-689 1:100) or APC isotype control (Miltenyi Biotech, 130-104-615 1:100) for 10 min at room temperature. Cells were analyzed using the BD Biosciences FACS LSR Fortessa X-20 instrument with at least 30,000 cells. Results were processed using FlowJo v10 (FlowJo, LLC).

## Human fibroblasts culture

Commercially available normal adult human dermal fibroblasts (NHDF-Ad Lonza CC-2511, Lot 545147) were used up to passage 6 to generate the tissues. They were cultured in T75 flasks, in DMEM high glucose, supplemented with 10% FBS, 1% NEAA, and 1% penicillin-streptomycin.

## Tissues generation and culture

The wells of the assay were first rinsed twice with PBS and incubated at 37°C in DMEM high glucose, 2.3 mM CaCl$_2$, 10%FBS, 0.1% penicillin-streptomycin. To generate the tissues, cardiomyocytes were dissociated by enzymatic digestion (Miltenyi Multi Tissue dissociation kit 3) on day 22 after differentiation. Fibroblasts were dissociated with TrypLE Express Enzyme (12605010 Thermo Fisher). Three ratios of hiPSC-CMs:fibroblasts were tested: 4:1, 3:1, and 2:1. For the following experiments the tissues are generated with a ratio of 3:1 hiPSC-CMs:fibroblasts as this ratio allowed the best preservation of tissues in time. A total of 150k cells were seeded in each well. The required numbers of fibroblasts and cardiomyocytes were then mixed and resuspended in 200 µL per well of DMEM high glucose, 2.3 mM CaCl$_2$, 10%FBS, 0.1% penicillin-streptomycin. The plate was then incubated at 37°C for 5 min and centrifuged three times in 'short' mode to ensure that the cells fall into the circular molds. The plate was then kept at 37°C and the medium was changed every 2 days.

## Tissues recording and analysis

The cardiac rings were recorded for 10 s in brightfield imaging with a high-speed CCD camera (PL-D672MU, Pixelink) mounted on a microscope (Primovert, Zeiss), at a ×10 magnification. The videos were then analyzed with a custom Matlab script (*Seguret, 2024*). Briefly, the area of the central pillar is monitored in time and allows to recover the contraction amplitude in time *Amplitude*(t)=*Max*(*Area*) – *Area*(t) and therefore the contraction strain in time (in terms of area) $\epsilon_A$ calculated as $\epsilon_A(t) = (Max(Area) - Area(t))/Max(Area)$ as well as the derivative of this strain in time. $\epsilon_A(t)$ is linked to the linear strain $\epsilon$ with the formula $\epsilon = \frac{\Delta r}{r_{max}} = \frac{\epsilon_A}{1+\sqrt{1-\epsilon_A}}$. As we measured the Young's modulus of the gel, we can recover the stress developed by the cardiac rings $\sigma = E.\epsilon(t)$. From these traces we also derive the contraction frequency by Fourier transform. It was considered that below a contraction of 0.5%, the signal-to-noise ratio was not sufficient enough to detect the contraction of the tissue and the beating frequency was therefore set to 0 Hz. Finally, the maximum contraction and relaxation speeds of the tissue were also calculated respectively from the maxima and minima of the derivative of this strain in time.

## Live imaging in tissues

Around 15 days after seeding, tissues were washed with a Tyrode solution (NaCl 14 mM, KCl 5 mM, 10 mM HEPES, glucose 10 mM, MgCl$_2$ 1 mM, CaCl$_2$ 1.8 mM, pH 7.4) and incubated 30 min at 37°C,

5% $CO_2$ with the FluoVolt Loading solution (FluoVolt dye dilution 1:1000, Powerload solution dilution 1:100 in the Tyrode solution) as per the manufacturer's recommendations (FluoVolt kit – Thermo Fisher). They were then washed with Tyrode and incubated in 200 μL Tyrode per well at 37°C, 5% $CO_2$ to recover. Live fluorescence imaging of the FluoVolt dye was performed in situ with a Leica TSC SP8 confocal imaging system in resonant mode at 37°C incubation, with a ×20 objective and a 256×256 pixels resolution. This allowed a frame rate of 43.96 fps.

### Response to increasing extracellular calcium concentration and drug tests

Tissues response to different drugs and increasing calcium concentrations was assessed after day 14. For the increasing extracellular calcium concentrations, tissues were first changed to Tyrode solution 0.5 mM in calcium (NaCl 140 mM, KCl 5 mM, 10 mM HEPES, glucose 10 mM, $MgCl_2$ 1 mM, $CaCl_2$ 0.5 mM, pH 7.4) and let to equilibrate for 30 min at 37°C, 5% $CO_2$. Calcium concentration was then increased by sequential addition of a Tyrode solution at 10 mM in calcium. After each addition, the tissues were incubated for 5 min before recording. For drug tests, tissues were changed to Tyrode solution at 1.8 mM in calcium (NaCl 14 mM, KCl 5 mM, 10 mM HEPES, glucose 10 mM, $MgCl_2$ 1 mM, $CaCl_2$ 1.8 mM, pH 7.4) and let to equilibrate for 30 min at 37°C, 5% $CO_2$. To evaluate the dose-response of the cardiac rings to the different drugs (verapamil – Sigma v0100000, isoproterenol Sigma 1351005, dofetilide – Sigma PZ0016), the contractility was first recorded for 30 s in basal state and the drug was then added sequentially. Between each addition of drug, the tissues were incubated for 5 min at 37°C before being recorded for 30 s.

### Immunofluorescence staining and confocal microscopy

Tissues were fixed at day 15 with 4% paraformaldehyde for 15 min and rinsed three times in PBS for 5 min. They were then blocked and permeabilized with blocking solution (BSA 2%, Triton 0.5%) at 4°C overnight. Primary antibodies vimentin diluted at 1:250 (MA5-11883, Thermo Fisher Scientific) and troponin T diluted at 1:500 (ab45932, Abcam) were added the next day and incubated at 4°C overnight. Tissues were rinsed three times in PBS for 5 min and secondary antibodies (Alexa Fluor secondary antibodies, Life Technologies, at 1:1000) and DAPI diluted at 1:1000 were added and incubated at 4°C overnight. Immunostaining pictures were taken with a Leica TCS SP8 confocal system.

### Statistical analysis

All numerical results are expressed as mean ± standard error mean (SEM) or standard deviation (SD) of three independent experiments (specified in for each experiment). Differences between experimental groups were analyzed with the appropriate statistical tests, specified each time. p-Values <0.05 were considered significant for all statistical tests. Statistical analyses were performed with GraphPad Prism software.

## Acknowledgements

We thank the iPS core facility of Paris Cardiovascular Research Center for its support for cell culture. We thank Camille Knops and Yunling Xu from the Flow Cytometry and the Microscopy platforms, respectively, from Université de Paris, Paris Cardiovascular Research Center, Paris, France.

## Additional information

### Competing interests

Patricia Davidson, Stijn Robben, Cyril Cerveau, Rita S Rodrigues Ribeiro: Employee of 4Dcell, which manufactured the 3D gel structures. The other authors declare that no competing interests exist.

### Funding

| Funder | Grant reference number | Author |
| --- | --- | --- |
| Leducq Foundation | 18CVD05 | Jean-Sébastien Hulot |

| Funder | Grant reference number | Author |
|--------|------------------------|--------|

The funders had no role in study design, data collection and interpretation, or the decision to submit the work for publication.

## Author contributions

Magali Seguret, Software, Formal analysis, Investigation, Writing – original draft; Patricia Davidson, Resources, Investigation, Writing – review and editing; Stijn Robben, Resources, Formal analysis, Writing – review and editing; Charlène Jouve, Validation, Investigation, Visualization, Project administration; Celine Pereira, Investigation, Methodology; Quitterie Lelong, Software, Investigation; Lucille Deshayes, Investigation; Cyril Cerveau, Resources, Validation, Methodology, Writing – review and editing; Maël Le Berre, Conceptualization, Resources, Funding acquisition, Writing – review and editing; Rita S Rodrigues Ribeiro, Resources, Supervision, Methodology, Writing – review and editing; Jean-Sébastien Hulot, Conceptualization, Data curation, Formal analysis, Supervision, Funding acquisition, Validation, Methodology, Writing – original draft, Project administration

## Author ORCIDs

Magali Seguret ⓘ https://orcid.org/0000-0002-3059-3113
Patricia Davidson ⓘ https://orcid.org/0000-0002-3939-3601
Charlène Jouve ⓘ https://orcid.org/0000-0002-5792-4621
Rita S Rodrigues Ribeiro ⓘ https://orcid.org/0000-0002-4139-7679
Jean-Sébastien Hulot ⓘ https://orcid.org/0000-0001-5463-6117

Reviewer #1 (Public review): https://doi.org/10.7554/eLife.87739.3.sa1
Author response https://doi.org/10.7554/eLife.87739.3.sa2

# Additional files

## Supplementary files

• MDAR checklist

## Data availability

The custom script for video imaging analysis has been deposited on Github (*Seguret, 2024*; copy archived at swh:1:rev:9eb2711291696f5c94eb709bdb82c819d426d7e1).

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
