## [Editor Report · eLife assessment]

This paper reports a **valuable** platform for cardiac tissue cultivation. The throughput, consistency of the tissue, and the potential integration of high-throughput automation are an advantage over other approaches. The tissues and the platform are validated using appropriate methodology to provide **convincing** evidence of the tissue cultivation capability.

---

## [Referee Report · Reviewer #1 (Public review)]

The manuscript, "A versatile high-throughput assay based on 3D ring-shaped cardiac tissues generated from human induced pluripotent stem cell-derived cardiomyocytes," developed a unique culture platform with PEG hydrogel that facilitates the in-situ measurement of contractile dynamics of the engineered cardiac rings. The authors optimized the tissue seeding conditions, demonstrated tissue morphology with expressions of cardiac and fibroblast markers, mathematically modeled the equation to derive contractile forces and other parameters based on imaging analysis, and concluded by testing several compounds with known cardiac responses.

The authors answered my questions with appropriate experiments and explanation.

(1) This paper presents an intriguing platform that creates miniature cardiac rings with merely thousands of cardiomyocytes per tissue in a 96-well plate format. The shape of the ring and the squeezing motion can recapitulate the contraction of the cardiac chamber to a certain degree. However, Thavandiran et al. (PNAS 2013) created a larger version of the cardiac ring and found that electrical propagation revealed spontaneous infinite loop-like cycles of activation propagation traversing the ring. This model was used to mimic a reentrant wave during arrhythmia. Therefore, there are concerns about whether a large number of cardiac tissues experience arrhythmia due to geometry-induced re-entry current and cannot be used as a healthy tissue model.

In the new experiment, the authors demonstrated with voltage-sensitive dye that these miniaturized tissues do not experience any arrhythmia, potentially due to their small size.

(2) The platform can produce 21 cardiac rings per well in 96-well plates, with the throughput being the highest among competing platforms. The resulting tissues exhibit good sarcomere striation due to the strain from the pillars. However, emerging questions pertain to culture longevity and reproducibility among tissues. According to Figure 1E, uneven ring formation around the pillar leads to tissue thinning and breakage. Only 50% survival is observed after 20 days of culture in the optimized seeding group. Are there any strategies to improve this survival rate? Additionally, do the cardiac rings detach from the glass slides and roll up, given the two compartments with cardiac and fibroblast-rich regions where fibroblasts maintain attachment to the glass slides? Moreover, the standard deviation of force measurement is a quarter of the value, which is suboptimal given the high replicate number. As the platform utilizes imaging analysis to derive contractile dynamics, calibration based on the angle and distance of the camera lens to individual tissues should be conducted to reduce error. On the other hand, how reproducible are the pillars? It is highly recommended to mechanically evaluate the consistency of the hydrogel-based pillars across different wells and within wells to understand the variance.

The authors stated that the platform has been tested and improved with multiple cell lines to enhance tissue survival rates. The methodology of image capture and calculation of contractile dynamics were explained in detail to address concerns. Moreover, the reproducibility of the pillars was demonstrated by consistent results of Young's Modulus (AFM) from each pillar, showing low standard deviations.

(3) Does the platform allow the observation of non-synchronized beating when testing with compounds? This can be extremely important as the intended applications of this platform are drug testing and cardiac disease modeling. The author should elaborate on the method in the manuscript and explain the obtained results in detail.

Referring to Question #1, the platform does not present arrythmia potentially due to the small size of the tissue.

(4) The results of drug testing are interesting. Isoperenoral is typically causing positive chronotropic and positive inotropic responses, where inotropic responses are difficult to obtain due to low tissue maturity. It is inconsistent with other reported results that cardiac rings do not exhibit increased beating frequency, but slightly increased forces only.

The authors repeated the experiment with the same results and hypothesized that the results would be line-dependent, since the maturation of iPSC-CM is not consistent. The additional dose curves provided more information on the tissue behaviors against well-known compounds.

Overall, the manuscript is well-written, and the designed platform presents unique advantages for high-throughput cardiac tissue culture. The paper has adequate data to demonstrate the proof-of-concept study of the platform. The throughput, consistency of the tissue, and the potential integration of high-throughput automation would be the highlights of this platform.

---

## [Author Response]

The following is the authors’ response to the original reviews.

**Recommendations for the authors:**

**Reviewer #1 (Recommendations For The Authors):**
(1) The author should evaluate the possibility of naturally occurring arrhythmia due to the geometry of the tissues, by using voltage or calcium dye.

Answer: We thank the reviewer for this suggestion. We have performed new experiments using a voltage-sensitive fluorescent dye (i.e. FluoVolt) with data reported in the new Figure 4 + new results section “arrhythmia analysis”. Briefly, we found that our ring-shaped tissues are compatible with live fluorescence imaging. We were then able to show that our cardiac tissues beat regularly, without naturally occurring arrhythmias or extra beats. We could not detect any re-entrant waves in our tissues in the conditions offered by the speed of our camera. A specific paragraph has also been added to the discussion.

(2) There is only 50% survival after 20 days of culture in the optimized seeding group. Is there any way to improve it? The tissues had two compartments, cardiac and fibroblast-rich regions, where fibroblasts are responsible for maintaining the attachment to the glass slides. Do the cardiac rings detach from the glass slides and roll up? The SD of the force measurement is a quarter of the value, which is not ideal with such a high replicate number.

Answer: This paper report seminal data that will serve as a foundation for further use of the platform. We are currently expanding to other cell lines with improvement in survival (see https://insight.jci.org/articles/view/161356). We confirm that the rings do not detach. The pillar was specifically designed to avoid this (See figure 1B).

As the platform utilizes imaging analysis to derive contractile dynamics, calibration should be done based on the angle and the distance of the camera lens to the individual tissues to reduce the error. On the other hand, how reproducible of the pillars? It is highly recommended to mechanically evaluate the consistency of the hydrogel-based pillars across different wells and within the wells to understand the variance.

Answer: We propose a system and a measurement method that do not need calibration. Contraction amplitude is expressed as a ratio between the contracted / relaxed areas (See figure 3 A). There is thus no influence of the distance of the camera lens.

In order to evaluate the consistency of the mechanical properties of the hydrogel, we reproduced the experiment pictured in Figure1-Supplement 1, and measured the Young’s Modulus of three different gel solutions on different days. In the three experiments performed, we found values of 10.0-12.2 kPa, resulting in a final average value of 11.2 (+/- 0.6) kPa, coherent with the value reported in the article. We are therefore confident that the mechanical properties are consistent across and within wells. More extensive mechanical characterization of the molded gels would require the access to an Atomic Force Microscope (AFM), and is considered in the future.

The author should address the longevity and reproducibility issues, by working on the calibration of camera lens position/distance to tissues and further optimizing the seeding conditions with hydrogels such as collagen or fibrin, and/or making sure the PEG gels have high reproducibility and consistency.

Answer: This paper report seminal data that will serve as a foundation for further use of the platform. This platform (including the design, approach and choice of polymers) allows a fast and reproducible formation of an important number of cardiac tissues (up to 21 per well in a 96-well format, meaning a potential total of about 2,000 tissues) with a limited number of cells.

(3) The evaluation of the arrhythmia should be more extensively explained and demonstrated.

Answer : See answer to comment 1

(4) The results of isoproterenol should be checked as non-paced tissues should have increased beating frequency with increasing dosages. Dofetilide does not typically have a negative inotropic effect on the tissues. Please check on the cell viability before and after dosing

Answer : We agree with this reviewer on the principle. However, we have repeated the experiments and we confirm our results, i.e. increasing concentrations of isoproterenol induced a trend towards increase in the contraction force and significantly increased contraction and relaxation speeds without change in the beat rate (Figure 5C). We do not have a definitive explanation for this observation. Our hypothesis is that this increase in contraction and relaxation speeds induced by isoproterenol is translated, on average in our study, into an increase in contractile force rather than in an increase in contraction frequency. This may depend on the cell line used, and is very well illustrated in a recent paper from Mannhardt and colleagues (Stem cell reports. 2020; 15(4):983–998). Of the 10 different cell lines tested in engineered heart tissues, all show an increase in contraction and relaxation speeds after isoproterenol administration, but this is translated either into an increase in contractile force (4 cell lines) or into a shortening of the beat (3 cell lines), and only 2 cell lines show an increase in both parameters. Indeed, since iPSC-CMs are immature cardiac cells, it is rare to obtain a positive force-frequency relationship without any maturation medium or mechanical or electrical training.We agree that above a concentration of 10nM, dofetilide shows cardiotoxicity in our tissues as tissues completely stop beating.

**Reviewer #2 (Recommendations For The Authors):**
In addition to the general comments in the public review, I have the following specific suggestions to the authors, that would help improve the manuscript.(1) Please describe the protocol for preparation of cardiac rings (shown in Figure 1C) in more detail. In particular, please describe how the tissues were transferred from the mold into the 96-well plate and how are they positioned and characterized during the study.

Answer: There is no transfer of the tissues as they directly form in the well, that is pre-equipped with the molded PEG gel (See Figure 1B and methods section). The in situ analysis is a strong asset of this platform.

(2) Please clarify the timepoints in this study. The overall schematic in Figure 1 C shows that the rings were formed on day 22 and then studied for 14 days, while Figure 2B shows data over 20 days following seeding, and Figure 3 shows data 14 days after seeding. It appears that these were separate studies optimization of myocyte/fibroblast ratio followed by the main study.

Answer: Figure 1C is showing the timeline including the cardiomyocytes differentiation. hiPSC-CMs are indeed seeded in the wells 22 days after starting the differentiation, which represent the Day0 for tissue formation. We apologize for the confusion.

(3) Please explain if the number of rings per well (Figure 2) was used as the only criterion for selecting the myocyte/fibroblast ratio, and if so, why. Were these rings also characterized for their structural and contractile properties?

Answer: Figure 2 supplement 1 report the contractility data according to the different tested ratios, and show no differences. The number for generated ring-shaped tissues was indeed the only criterion retained.

(4) Please provide rationale for using the dermal rather than cardiac fibroblasts.

Answer: We had previous experience generating EHTs using dermal fibroblasts which are easier to obtain commercially. Our approach could in theory also work using cardiac fibroblasts, which we have not tested in the present study.

(5) Figure 2 panels C-E show an interesting segregation of cardiomyocytes into a thin cylindrical layer that does not appear to contain fibroblasts and a shorter and thicker cylinder containing fibroblasts mixed with occasional myocytes. Please specify at which time point this structure forms, and how does it change over time in culture? At which time point were the images taken? It would be helpful to include serial images taken over 1-14 days of study.

Answer: We thank the reviewer for this interesting comment. We have performed additional immunostainings (reported in Figure 2 supplement 3) on tissues at Day 1 and day 7 after seeding. The segregation appears in the 7 first days. It appears that 1 day after seeding the fibroblasts are not yet attached, although the cardiac fiber has already started to be formed. Seven days after seeding, fibroblasts are fully spread and attached, and the contractile ring is formed and well-aligned. Brightfield images are reported in Figure 1E.

(6) In the cardiomyocyte region (Figure 2D) the cells staining for troponin seem to be only at the surfaces. The thickness of the layer is only about 30-40 µµ, so one would assume that cell viability was not an issue. Please specify and discuss the composition of this region.

Answer: We agree but we think this is a technical issue as at the center of the tissue, tissue thickness will limit laser penetration, although at the surface (inner our outer), the laser infiltrates easily between the tissue and the PEG. Moreover, we see on the zoomed view of the tissue in Figure 2 Supplement 2 that we have a staining inside the cardiac fiber, which just appears less strong due to tissue thickness.

(7) Please also discuss segregation in terms of possible causes and the implications of apparently very limited contact between the two cell types, i.e., how representative is this two-region morphology of native heart tissue. Also, it would be interesting to know how the segregation has changed with the change in myocyte/fibroblast ratio.

Answer: We are not sure there is a very limited contact as the use of fibroblasts is critical to ensure the formation of tissues (i.e. no tissues can be formed if we avoid the use of fibroblasts). We agree that these ring-shaped cardiac tissues are not especially representative of a native heart tissue in terms of interactions between several cell types. They were developed as a surrogate for physiopathological and pharmacological experiments (see a recent application in https://insight.jci.org/articles/view/161356)

(8) There is interest and demonstrated ability to culture engineered cardiac tissues over longer periods of time. Please comment what was the rationale for selecting 14-day culture and if the system allows longer culture durations.

Answer: In line with this comment, we have studied the contractile parameters of our rings 28 days after seeding and compared to their contractile parameters at D14. We found a slight increase for all the parameters, which is significant for the maximum contraction speed. Nevertheless, the data is much more variable and the number of tissues is lower (29 for D14 against 17 for D28). Therefore, we demonstrated that long-term culture of our tissues is possible, however not yet optimized. Hence, the following physiological and pharmacological tests have been done at D14.

(9) Figure 3 documents the development of contractile parameters over 14 days of culture. Would it be possible to replace the arbitrary units with the actual values? Also, would it be possible to include the corresponding images of the rings taken at the same time points, to show the associated changes in ring morphologies.

Answer: Contraction amplitude is expressed as a ratio between the contracted / relaxed areas (See figure 3 A): it is a ratio, thus without unit. Corresponding images can be seen in Figure 1 E.

(10) The measured contraction stress, strain, and the speeds of contraction and relaxation improve from day 1 to day 7 and then plateau Figure 3, Supplemental Figure 3. Please discuss this result.

Answer: The new immunostainings performed on tissues at Day 1 and Day 7 show the progressive alignment of the cardiomyocytes and the muscular fibers, with an almost complete organization at Day 7.

(11) The beating frequency does not appear to markedly change over time, while Figure 3B shows strong statistical significance (***) throughout the 14-day period. Please check/confirm.

Answer: We confirm this result.

(12) Please comment on the lack of effect of isoproterenol on beating frequency.

Answer: We agree with this reviewer on the principle. However, we have repeated the experiments and we confirm our results, i.e. increasing concentrations of isoproterenol induced a trend towards increase in the contraction force and significantly increased contraction and relaxation speeds without change in the beat rate (Figure 5C). We do not have a definitive explanation for this observation. Our hypothesis is that this increase in contraction and relaxation speeds induced by isoproterenol is translated, on average in our study, into an increase in contractile force rather than in an increase in contraction frequency. This may depend on the cell line used, and is very well illustrated in a recent paper from Mannhardt and colleagues (Stem cell reports. 2020; 15(4):983–998). Of the 10 different cell lines tested in engineered heart tissues, all show an increase in contraction and relaxation speeds after isoproterenol administration, but this is translated either into an increase in contractile force (4 cell lines) or into a shortening of the beat (3 cell lines), and only 2 cell lines show an increase in both parameters. Indeed, since iPSC-CMs are immature cardiac cells, it is rare to obtain a positive force-frequency relationship without any maturation medium or mechanical or electrical training.

(13) Please compare the contractile function of cardiac tissues measured in this study with data reported for other iPSC-derived tissue models.

Answer : A specific paragraph tackles this aspect in the discussion